Real-time bioacoustics monitoring and automated species identification

Aide T. Mitchell 1 tmaide@yahoo.com
Corrada-Bravo Carlos 2
Campos-Cerqueira Marconi 1
Milan Carlos 1
Vega Giovany 2
Alvarez Rafael 2
1 Department of Biology, University of Puerto Rico-Rio Piedras , San Juan, Puerto Rico , United States
2 Department of Computer Science, University of Puerto Rico - Rio Piedras , San Juan, Puerto Rico , United States
Huang Xiaolei
Electronic publication date: 2013 Jul 16
Publication date: 2013
Volume: 1
Electronic Location ID: e103
Received 2013 May 7; Accepted 2013 Jun 22
Copyright: © 2013 Aide et al.
Copyright year: 2013
Copyright holder: Aide et al.
License: This is an open access article distributed under the terms of the Creative Commons Attribution License, which permits unrestricted use, distribution, and reproduction in any medium, provided the original author and source are credited.
License URL: https://creativecommons.org/licenses/by/3.0/

Keywords: Acoustic monitoring, Machine learning, Animal vocalization, Long-term monitoring, Species-specific algorithms

Funding: DOD Legacy program W912DY-07-2-0006- P00001, P00002, P0003 National Science Foundation 0640143 University of Puerto Rico-Rio Piedras FIPI Funds for this research were provided by the DOD Legacy program (W912DY-07-2-0006- P00001, P00002, P0003), National Science Foundation (0640143), and the University of Puerto Rico-Rio Piedras (FIPI). The funders had no role in study design, data collection and analysis, decision to publish, or preparation of the manuscript.

==============================
Traditionally, animal species diversity and abundance is assessed using a variety of methods that are generally costly, limited in space and time, and most importantly, they rarely include a permanent record. Given the urgency of climate change and the loss of habitat, it is vital that we use new technologies to improve and expand global biodiversity monitoring to thousands of sites around the world. In this article, we describe the acoustical component of the Automated Remote Biodiversity Monitoring Network (ARBIMON), a novel combination of hardware and software for automating data acquisition, data management, and species identification based on audio recordings. The major components of the cyberinfrastructure include: a solar powered remote monitoring station that sends 1-min recordings every 10 min to a base station, which relays the recordings in real-time to the project server, where the recordings are processed and uploaded to the project website (arbimon.net). Along with a module for viewing, listening, and annotating recordings, the website includes a species identification interface to help users create machine learning algorithms to automate species identification. To demonstrate the system we present data on the vocal activity patterns of birds, frogs, insects, and mammals from Puerto Rico and Costa Rica.

Introduction

Ecologists, conservation biologists, and park and resource managers are expected to make decisions to mitigate or manage the threats of climate change and the high rates of species loss. Unfortunately, they rarely have the information needed to make informed decisions because our understanding of most biological systems is based on very limited spatial and temporal coverage. In most biomes, data collection, particularly of the fauna, is concentrated in a few sites, and this highly aggregated distribution of information, limits our ability to understand large-scale ecological processes and to properly manage fauna in large areas (Gentry, 1990; Terborgh et al., 1990; Condit, 1995; Porter et al., 2005; Underwood, Hambäck & Inouye, 2005; Porter et al., 2009). Furthermore, long-term information is needed to understand the implications of land and climate change on biological systems (Porter et al., 2005). From both a conceptual and management perspective there is an urgent challenge to increase biological data collection over large areas and through time.

What is needed are long-term population and distribution data for thousands of species across their range. For some economically important species (e.g., salmon) we have long-term data (Niemela, Julkunen & Erkinaro, 2000), but for the majority of species the data is limited to a few years and a few populations. Other areas of science, such as meteorology and land change science have taken advantage of new technologies, such as inexpensive sensors, wireless communication, and satellite images to expand their data sets to the global scale (Porter et al., 2009). Given the urgency of the biodiversity crisis, it is essential that we take advantage of all available tools to improve biodiversity monitoring to thousands of sites around the world.

Traditionally, biodiversity is assessed using a variety of methods that are generally costly, limited in space and time (e.g., Parker, 1991; Sauer, Peterjohn & Link, 1994; Sueur et al., 2008), and most importantly, they rarely include a permanent record. Furthermore, most fauna monitoring protocols require the presence of experts in the field because data are often acquired through indirect cues (e.g., animal vocalizations). This creates various problems. First, in terms of acoustic identification, there are few experts that can confidently identify animals based on vocalization, yet there are many studies that could benefit from this information. Second, experts vary in their abilities to correctly identify species, and this leads to observer bias (Fitzpatrick et al., 2009). Additionally, these protocols often collect data over a very limited spatial and temporal scale, and these constraints reduce the researcher’s ability to understand the dynamic patterns of animal populations. Furthermore, most traditional sampling methodologies do not include a permanent record and, thus, there is no way to validate the data.

In contrast, automated digital recording systems can monitor animal populations 24 h a day, every day of the year, in stations across a variety of habitats simultaneously, and all recordings can be permanently stored (Acevedo & Villanueva-Rivera, 2006; Brandes, 2008; Lammers et al., 2008; Sueur et al., 2008; Acevedo et al., 2009; Hoeke et al., 2009; Tricas & Boyle, 2009). This type of monitoring can be effective because in most ecosystems a large proportion of the fauna emits sounds for a variety of reasons including inter- and intraspecific communication, orientation (Peter & Slabbekoorn, 2004), and detection and localization of prey and predators (Richardson et al., 1995), but most importantly, these sounds are species specific.

Automated data collection systems can collect an overwhelming amount of data, creating problems with data management and analysis (Villanueva-Rivera & Pijanowski, 2012). To help solve these problems, researchers have developed algorithms to automate species identification of vocalizations of bats (Herr, Klomp & Atkinson, 1997; Walters et al., 2012; Parsons & Jones, 2000), whales (Murray, Mercado & Roitblat, 1998; Brandes, 2008; Marques et al., 2012; Mellinger & Clark, 2000; Moore et al., 2006), dolphins (Oswald, Barlow & Norris, 2003), insects (Chesmore, 2004; Chesmore & Ohya, 2004), and birds and amphibians (Anderson, Dave & Margoliash, 1996; Kogan & Margoliash, 1998; Acevedo & Villanueva-Rivera, 2006; Hilje & Aide, 2012; Ospina et al., 2013). A limitation with this approach is that most users do not have the programming or math skills to develop these algorithms. Furthermore, most projects have only produced algorithms for one or a few target species.

In this manuscript, we describe the acoustical component of the Automated Remote Biodiversity Monitoring Network (ARBIMON), a novel combination of hardware and software (cyberinfrastructure) for automating data acquisition, data management, and identification of multiple species of amphibians, birds, insects, and mammals. The main objectives of the manuscript are to demonstrate: (1) how detailed, long-term acoustical data can be collected and managed, (2) how users can create species-specific identification algorithms with no machine learning experience, and (3) how the information created by the system can be used to better understand the activity patterns and long-term population trends of the fauna. To demonstrate this system we present data on the activity patterns of nine species (4-amphibians, 2-birds, 1-mammals, and 2-insects) from an herbaceous wetland in Puerto Rico and a lowland tropical forest in La Selva Biological Station in Costa Rica.

Methods

Data acquisition

The cyberinfrastructure for collecting and storing the audio recordings includes: (1) the acoustic permanent station, (2) the field base station, and (3) the ARBIMON server (Fig. 1). The permanent monitoring station includes an iPod Touch (2G) with a pre-amplifier, which is powered with a 50 W solar panel, voltage converter, a router, and a 12 V car battery (Fig. 1). A microphone with a frequency response range from 20 Hz to 20 kHz is attached to the iPod via the pre-amplifier. The battery, pre-amplifier, voltage converter, router, and iPod are housed in a water/shock proof case. The pre-amplifier has three gain settings. The gain was set at the intermediate level. Informal experiments suggest that this recording systems will detect the common coqui (Eleutherodactylus coqui) in a forest habitat up to approximately 50 m, suggesting that for this species the sampling area would be approximately 1 ha. An application in the iPod controls the duration of the recording and the time between recordings. Presently, it is programmed to record 1 min of audio every 10 min for a total of 144 1-min recordings per day. The recording schedule can be easily modified depending on the objectives of each project. The application generates a filename for each recording, instructs the software to make the recording, and sends the recording using Secure Copy (SCP) to a MacMini computer at the base station. These files are forwarded by wireless communication from the iPod to a router that is connected to a directional antenna (Avalan Wireless 900 MHz Radio Ethernet extender), which forwards the file to the receiving antenna that is connected to the base station computer. Our experience shows that this radio/antenna system can maintain a strong connection at a distance of 2 km through vegetation and up to 40 km if there is line-of-sight between the antennas.

Figure 1 Workflow of data acquisition, processing, and management.

The main functions of the base station are to provide internet access, store all data files locally on a 1Tb external hard drive, compress the recordings to reduce the file size, and to forward these files to the project server at the University of Puerto Rico (Fig. 1). These functions are activated every time a recording is received via a folder action and an Applescript. The script converts the recording from stereo to mono, and compresses it using flac format (an open source alternative for lossless compression and decompression of audio files, http://flac.sourceforge.net/), stores the file locally, and sends a copy to the project server. The project server, an Apple Xserve (2.8 GHz Quad-Core Intel Xeon, 4–12 GB 800 MHz DDR2 FB-DIMM) running MacOS X 10.5.4 Server, Apache 2.2, Php 5.2.5 and MySQL5.0.45, is used for data storage, data backup, data management, analysis, and web hosting. The server also includes a Promise VTrak E610f RAID Subsystem with 12TB configured as a RAID6 for a total of 9TB of available space.

In addition, acoustic files collected using portable recorders (e.g., Passive Acoustic Monitoring (PAM) equipment) can be uploaded to the database. These files are managed and analyzed in the same way as the recordings from the acoustic permanent stations.

Database and data management

A normalized open source database schema using the MySQL database system is the cornerstone of our web application. The database is general enough so that it can be used for any acoustic project, allowing researchers to work with the data of their specific projects, but when appropriate it allows the merger and sharing of data among projects.

The centerpieces of the design are the sensors that acquire the data and the methods used to process the data, allowing our system to handle a variety of sensors, use different configurations of these sensors, and to create an efficient way to relate the data with the type of sensor and configuration. Additionally, this database architecture provides easy access to the data at different points in the processing path. This was accomplished by handling the data as both input and output, thus each data entity is output in one instance and input in the other. Up to now the principal sensors have been the recording stations described above and the core data of the database are the audio recordings (Fig. 1) with their associated attributes: recording site (id, name, longitude, latitude and elevation) and study area (id, name, organization in charge and time zone).

Database management

Although anyone can view and listen to the recordings on the project website, only approved users can analyze or annotate recordings. To manage projects and users within projects we have developed an administrative interface, which has three sections: administration, project creation and management, and global security. The administration component maintains the databases of all projects, keeps a log of all users’ activity, and documents any security breach or system failure. The project creation and management component allows a new user to (1) create a project, (2) specify site names, location, and time zone, and (3) assign users with different privilege levels of to the project. The global security component manages users and their privileges.

Data processing

When the audio files arrive to the project server, they are archived, and then sent to a program that extracts the raw data from the wav format to create a spectrogram of the recording. This spectrogram is created using a short-time Fourier transform (STFT) using 512 samples and a Hann window overlapping 256 samples. For one-minute recordings with a sample rate of 44,100 samples per second each cell of the matrix represents 86 Hz by 0.005 s. This matrix is used to generate the spectrogram image of the recording and is the input for another program that demarks areas of high energy within the recording as regions of interest (ROIs). In addition, an mp3 file is generated using LAME (http://lame.sourceforge.net/) a high quality MPEG Audio Layer III (MP3) encoder licensed under the Lesser General Public License (LGPL). The smaller size of mp3 files makes them more appropriate for the web application, but the quality of the spectrogram or ROIs are not affected because they were generated using the original wav files.

The algorithm to create the regions of interest (ROIs) starts by analyzing the frequency-time matrix to determine the level of background noise within each frequency band. This information is used to define thresholds of audio intensity that the input signals in the recording must surpass to be considered as an acoustic event. For each frequency band, we determine the mean intensity value and keep only the samples that are greater than 10% above the mean. This process greatly reduces the data, making it suitable for storing as a compressed sparse matrix (CSR). We analyze the CSR containing the acoustic events using a depth-first search algorithm to create neighborhoods of pixels into a single region of interest (ROI). Once, the sample is used in a ROI they are removed from the CSR and the algorithm selects another event until all samples that were selected as an acoustic event participates in a ROI. The time and frequency variables that describe the bounding box of each ROI (minimum and maximum frequency, duration, maximum intensity and bandwidth) are the variables that are later used to create the automated species identification algorithms.

User interface for automating species identification

To automate species detection, we developed an application that uses Hidden Markov Models (HMMs). The application was designed so that the users can develop their own models using tools to view and listen to their recordings and to create, test, and validate species-specific identification models. The four major components that make up this interface included: (1) visualizer, (2) species validation, (3) model builder, and (4) model application.

Visualizer

This module is used for viewing, listening, and annotating recordings. The visualizer was developed in OpenLaszlo (a flash framework) so that it would be compatibility across browsers. The interface can accept recordings of any length and from most recording devices. The visualizer includes tools/features (e.g., zoom, filters) to facilitate viewing, listening, and data analysis.

Species validation

This tool allows the user to specify which species/vocalization is present or absent in each recording (Fig. 2). Users need to have a validation data set to verify the accuracy and precision of each model. In addition, the user can determine if the particular vocalization is correctly marked by the automated ROI generator.

Figure 2 The ARBIMON-acoustic web-based tools for creating, testing, and applying the species-specific identification models.

Model builder

This component has four sub-components. a. Training data – The first step in developing a species-specific model is to provide training data for the model (Fig. 2). The user provides the training data by identifying examples of the vocalization. Each model is based on a specific vocalization of a species. The user selects a series of ROIs from the recording that reflect the desired vocalization model. For example, two chirps followed by a shrill. This process is repeated to provide the program with additional training examples. This information is saved in the database and is later used for the optimization of the model using the Baum-Welch algorithm (Baum et al., 1970).

b. Model creation – We describe the sequence of a song as a Hidden Markov Model (HMM). The model is expressed as λ = (A, B, π) where A is a probability matrix for the transitions between states, B is a probability matrix for the emissions given the state and π is a vector of the probabilities of each state in the sequence. These probabilities are then optimized based on the observations in the training set using the Baum-Welch algorithm. The application requires the user to define the number and types of tones/notes in the species vocalization (Fig. 2). Then, using the training data acquired by the users, the program calculates the initial probabilities for the transition and emission matrices. The result of the Baum-Welch algorithm are the three optimized matrices A′, B′, π′ that are then used to calculate the probability that a given observation was generated by the model λ.

c. Applying model – The initial model can be applied to any number of recordings (e.g., the default is 500 random recordings) in the database. The web application allows the user to visualize the results of the initial model, select correct responses, incorporate the correct responses into the training data to improve the model, and then reanalyze the data if necessary. These tools and the iterative process quickly allow the user to build an accurate species identification model. Once the user is satisfied with the model, it can then be tested against the validation data.

d. Validation – In this step, the system applies the model only to the recordings that were validated for the presence/absence of the species/vocalization (Fig. 2). Next, the user is provided with an error matrix and statistics on the accuracy and precision of the model. Based on these statistics the user can modify the model by varying the range of values (e.g., minimum frequency, duration) used in determining which ROIs are used in the model. In addition, in this component the user can review the results. For example, the user can inspect recordings with false positives to determine how to improve the model.

The error or confusion matrix shows the number of true positives (species/vocalization determined as present by the user and detected by algorithm), true negative (species/vocalization determined as absent by the user and not detected by the algorithm), false positives (species/vocalization determined as absent by the user, but detected by the algorithm) and false negatives (species/vocalization determined as present by the user, but was not detected by the algorithm). In addition, the output includes estimates of precision and accuracy, which are calculated as: (1) Precision = true positives/ (true positives + false positives)

(2) Accuracy = (true positives + true negatives) / total

Model application

In this component, the user can apply the model to their complete data set (Fig. 2). In our case, we have tested the system with more than five years of 1-min recordings (n = 173,526) from our original permanent recording station site in Sabana Seca, Puerto Rico, and 19,043 recordings from La Selva Biological Station in Costa Rica. The system took less than two hours to run the three models for Sabana Seca through all of the recordings. The results from this analysis can be exported in cvs format for further analyses. In addition, the user can “publish” the model, making it available to other users and other projects.

Study site and study species

To demonstrate the use of the ARBIMON-acoustic application we created species-specific models for amphibians, birds, mammals, and insects based on recordings from a site in Puerto Rico and a site in Costa Rica. The species were selected to cover a range of taxa with different types of vocalizations. Vocalizations of frogs and birds were confidently identified based on our experience and comparisons with different sources of animal calls. Unfortunately, the two insect species, most likely cicadas, could not be captured and identified, but we carefully documented the call characteristics to assure that we modeled a specific species in each site.

The site in Puerto Rico, Sabana Seca (SS), is a small (180 ha) wetland near the Caribbean Primate Research Center (CPRC) in Toa Baja, Puerto Rico (18°25′56.01″ N and 66°11′45.62″ W). Typha dominguensis (cattail) is the dominant species in the wetland. This site is the only known locality of Eleutherodactylus juanariveroi (coqui llanero), an endangered frog species that was recently discovered (Rios-Lopez & Thomas, 2007). The major motivation for establishing a permanent recording station in Sabana Seca was to improve the information on the calling activity and population dynamics of E. juanariveroi. The station was established in March 2008, and for this study we present the results of species-specific identification models of the endemic frog species, E. juanariveroi, an exotic frog species Rana gryllo (pig frog), and an unidentified insect (insect #1).

The other study site was La Selva Biological Station (LSBS) in Costa Rica (10°25′ N, 84°01′ W). This reserve encompasses approximately 1,510 ha of which 64% is primary tropical forest, and contains a high diversity of flora and fauna (Clark & Gentry, 1991). The objective of this project was to conduct broad acoustic monitoring within mature forest for all species that contribute to the acoustic community. For this site, we created species-specific identification models for six species: Tinamus major (great tinamou), Ramphastos swainsonii (chestnut-mandibled toucan), Oophaga pumilio (strawberry poison-dart frog), Diasporus diastema (tink frog), Alouatta palliata (mantled howler monkey), and an unidentified insect (insect #2).

In addition to the recordings from the two permanent stations described in this manuscript, other recordings have been added to the ARBIMON database from other permanent stations in Puerto Rico, Hawaii, and Arizona, and from portable recording systems in Puerto Rico, Costa Rica, Argentina, and Brazil. As of May 7, 2013, the system has >1.3 million 1-min recordings, which can be freely accessed through the project web page (arbimon.net).

Results

Species identification models

To determine the accuracy and precision of the species identification models we compared the decisions made by the expert (i.e., validation data set) with the decision made by the models (Table 1). The Oophaga pumilio vocalization model had the highest accuracy (99%), while the model for insect sp#2 had the lowest accuracy (79%). Similarly, the Oophaga pumilio vocalization model had the highest precision (100%), but the Alouatta palliata model had the lowest precision (76%) due to the high level of false positives. In general, most of the models had relatively low levels of false positives (<5%), and higher levels of false negatives. For example, the Tinamous major model reported only 1 presence when the vocalization was actually absent (i.e., false positive), but 41 times the model reported the species was absent when it was really present (i.e., false negative). These results suggest that these models are relatively conservative; they rarely confused the species with another, but they do not always detect the species when it is present as determined by an expert through visual and/or aural inspection.

Table 1 Confusion matrix of the species-specific models.

The confusion matrix results based on a comparison of the validation training set for each of the nine species with the model results.

Species	Site	Validation
data (n)	True
positives	False
positives	True
negatives	False
negatives	Accuracy	Precision	
Oophaga pumilio	LSBS	183	31	0	150	2	99	100	
Ramphastos swainsonii	LSBS	395	24	5	348	18	94	83	
Alouatta palliata	LSBS	342	35	11	288	8	94	76	
Tinamus major	LSBS	407	67	1	298	41	90	99	
Rana grylio	SS	127	37	6	76	8	89	86	
Eleutherodactylus juanariveroi	SS	231	109	6	88	28	85	95	
Insect 01	SS	130	50	7	61	12	85	88	
Diaspora diastema	LSBS	190	54	4	101	31	82	93	
Insect 02	LSBS	163	53	1	75	34	79	98	
Notes.

LSBS – La Selva Biological Station, Costa Rica; SS – Sabana Seca, Puerto Rico.

There are two main causes for the false negatives. First, if the ROI generator does not mark the vocalization, it will not be incorporated into the analysis. This usually happens when the calling individual is far from the microphone and the vocalization was too faint to be detected by the ROI generator, but the expert could observe or hear the species in spectrogram and included the species as present in the validation data set. A second cause of false negatives occurred because we restricted the range of some parameters to minimize false positives, which could increase the number of false negatives.

There were many different causes of false positives. For example, thunderstorms created ROIs that were similar to those of Alouatta palliata. Mechanical noise caused by wind was the main cause of misidentifications of Rana grylio. The main source of false positives of Diaspora diastema was vocalizations of Oophaga pumilio. Nevertheless, this level of confusion in the identifications of D. diastema did not significantly change the description of the daily vocal activity pattern in comparison with previous studies (Graves, 1999; Hilje & Aide, 2012).

Species daily and annual activity patterns

These species-specific models were applied to all recordings from the two sites (SS – 173,526; LSBS – 19,043), and the detection data were used to determine the patterns of daily (SS and LSBS) and annual (SS) activity.

In Sabana Seca, the vocalization patterns of the three species were concentrated during the night, but the peak in activity of each species occurred at different times (Figs. 3A–3C). The native species, Eleutherodactylus juanariveroi had two peaks of vocal activity, one at dawn (5:00) and a higher and narrower peak at dusk (18:00). The exotic frog, Rana grylio, had a peak of vocal activity at 4:00; while insect sp #1 had a peak of activity at 21:00. The two frog species had low levels of activity during the day (6:00–18:00), and there were virtually no detections of the insect during the day.

Figure 3 Vocal activity in Sabana Seca.

Daily (A–C) and monthly (D–F) vocal activity of three species from Sabana Seca, Puerto Rico. The number in parenthesis is the number of recordings where the species was detected by the model. The detection frequency was calculated as the number of recordings with a positive detection divided by the total number of recordings during the time period.

The same data were used to visualize the pattern of vocal activity between October 2008 and April 2013 (Figs. 3D–3F). On average, the monthly detection frequencies of E. juanariveroi were around 0.20, but between October 2008 and May 2012 there was a significant decline in vocal activity (Ospina et al., 2013). Our data show that since May 2012 there has been a dramatic increase in detection frequency, and in September 2012, E. juanariveroi was detected in ∼30% of the recordings. The activity pattern of Rana grylio was more seasonally predictable. Each year there was a peak in vocal activity during the rainy season, between April and October, when calling activity (i.e., detection frequency) increased from <0.02 during most of the year to ∼0.10 during the peaks. In 2009, the detection frequency increased to 0.30 during the peak. These results reflect the biology of this aquatic species, which breeds during the wettest and warmest time of the year (Thorson & Svihla, 1943). In contrast to the seasonal pattern of R. grylio, the vocal activity of insect sp#1 was highly variable and much less frequent (Fig. 3F). In some months the species was rarely detected, but the following month the detection rate could increase by 2 to 4 fold, suggesting that the population of this species is highly dynamic.

In La Selva Biological Station, the variable pattern of daily vocal activity reflects the diversity of taxa that were studied (Fig. 4). The great tinamou (Tinamus major) and the chestnut-mandibled toucan (Ramphastos swainsonii) had peaks of activity at dawn and another at dusk, as is expected for most bird species (Terborgh et al., 1990). The howler monkey (Alouatta palliata) also had peaks of activity at dawn and dusk, but in contrast with the two bird species, it had a larger proportion of its detections during the day. The two frog species had very contrasting daily patterns of vocalization (Figs. 4D–4E). The peak in activity of Diaspora diastema occurred during the night with a peak of activity at 3:00 and small peak at 18:00, but there was also a low level of activity throughout the day. In contrast, the majority of vocal activity of Oophaga pumilio occurred during the day, with a peak (>28% of detections) at 7 am. The model for insect #2 showed virtually no activity during the day and a peak in vocalization around 22:00.

Figure 4 Vocal activity in La Selva.

Daily vocal activity of six species from La Selva Biological Station, Costa Rica. The number in parenthesis is the number of recordings where the species was detected by the model. The detection frequency was calculated as the number of recordings with a positive detection divided by the total number of recordings during the time period.

Discussion

How detailed, long-term acoustical data can be collected and managed

Here we have demonstrated how frequent (sub-hourly) data collection over long time periods (years) can be carried out, and how the data can be managed, archived, and analyzed virtually in real-time. By recording one minute of audio, every 10 min, we were able to achieve fine temporal resolution, covering 24 h a day, seven days of the week over a five year period in Puerto Rico. This fine-scale and long-term temporal sampling, now needs to be matched spatially with many sensors across the landscape.

The detailed and long-term temporal sampling of these sites could not have been accomplished without automating data acquisition, processing, and management. The automation of data collection also provided additional benefits. First, recordings can be inspected visually and aurally in real-time. Recordings from the Sabana Seca station took less than 1 min to be sent from the field, to the base station, and on to the project server where it was processed, stored, and incorporated into the project’s open-access web site. This real-time monitoring can help researchers and managers respond rapidly to important events, particularly when a model that identifies a focal species has been incorporated into the data processing scripts. Another benefit of the real time processing is that we can easily detect any malfunction of the hardware or software by inspecting the recordings, and then respond quickly to limit data loss. The Sabana Seca system collected recordings between 60 and 70% of the time. The major causes of data loss were: (1) loss of power due to extended cloud cover or vegetation growing over the solar panel, (2) loss of power at the base station, and (3) network problems at the base station. Nevertheless, the real cause of missing data was a slow response by our staff to solve these problems. To accelerate the response time, we have developed an application that continuously collects information from each station and generates an alert in the form of an email to the project owner when the station is malfunctioning.

Other benefits of automating data collection include: (1) reduced observer bias and (2) each recording is a verifiable permanent record, equivalent to a museum specimen. Even if observers could stay in the field 24 hr/d throughout the year, there would still be a problem of observer bias (Cerqueira et al., in press). This is a major limitation especially when it is necessary to sample many sites simultaneously or when data are collected over many years by many different observers. The ability to detect and identify an animal vocalization correctly may require years of experience. But, there can also be high levels of variation among “experts” due to differences in the habitat being sampled, hearing ability, or biases toward certain species (Sauer, Peterjohn & Link, 1994). Another benefit is that each recording is a permanent record, which allows multiple users to review them, leading to more accurate identifications and consequently more accurate estimates of population parameters. All recordings archived in ARBIMON (arbimon.net) are open access, and thus it is the equivalent of an acoustic museum, presently with >1.3 million 1-min recordings.

Our approach is very different in comparison with most other collections of animal vocalizations. For example, the recordings from the Macaulay Library of the Cornell Lab of Ornithology, Xeno-canto, and the Internet Bird Collection are important collections of animal vocalizations and photographs, but their focus is species specific. Furthermore, many species are represented by one or a few recordings. In contrast, our approach is to record the environment (i.e., soundscape), frequently and over the long-term. This allows multiple users to take advantage of the recordings. For example, while the initial objective of a project may be to study a specific bird species, the vocalizations of many other species (e.g., insects, frogs, birds and mammals) are likely to be present. In addition, a soundscape index, an integrated measure of the acoustic environment, can be calculated and measured across time to estimate changes in biodiversity or other factors affecting the acoustic environment (Sueur et al., 2008; Pijanowski et al., 2011). Moreover, given that all recordings will be permanently archived, future users, with new tools and questions will be able to reanalyze these recordings in the future.

Although there are many benefits of a permanent station, the user must consider the costs and other limitations. The initial cost of establishing a permanent recording stations will vary depending on the site and logistics, and could range from approximately US $10,000 to $20,000. Another important cost is the processing and long-term storage of the audio files. We have estimated the cost at US $0.15 per 1-min recording. Other limitations associated with any monitoring program that depends on audio recordings include: (1) poor or no detection of species or individuals that rarely use acoustic signals for communication (e.g., females and juvenile), (2) a single permanent or fixed station will only record biotic activity in a limited radius around the station and this distance will vary among species depending on the sound pressure generated by the calling individual (Llusia, Marquez & Bowker, 2011), and (3) using models to identify species-specific vocalizations in recordings with varying degrees of intense background noise (e.g., other species, rain, wind, automobile traffic) could result in misidentifications.

Species-specific identification models and daily and long-term activity patterns

For many studies, presence/absence data or an index of relative abundance can be very useful, but it is not easy to extract this information from thousands of recordings. While some researchers have the programming skills to manage and analyze their recordings, most do not. Typically, researchers resort to listening to a subset of their recordings, which can be very time consuming and leads to a considerable loss of data. In contrast, the ARBIMON-acoustic software allows the user to reduce the time analyzing recordings, while taking advantage of the complete data set. To do this the user must only inspect a subset of the recordings to provide examples of the species-specific vocalization (i.e., training data) and create the validation data set, which is needed for training the initial model and to evaluate the accuracy and precision of each model, respectively. Our results illustrate that the species-specific identification models created using the ARBIMON-acoustic system worked well for birds, mammals, amphibians and insects, and the models had high levels of accuracy and precision. These models allowed us to process 100,000s of recordings to generate detailed information on daily and monthly vocalization patterns for these species. Another important feature is that these models can be used in other projects, allowing new users to dedicate their time to producing new models of other vocalizations made by the same species or of other species. Most importantly, these web-based tools greatly simplify the process of extracting useful results for researchers and managers from the raw data (i.e., recordings), which should help the users to improve and expand their ecological monitoring programs.

We thank the past and present members of the ARBIMON team for their support in developing and maintaining the hardware and software. We thank the personnel of the Caribbean Primate Research Center and La Selva Biological Station (LSBS) in Costa Rica for their logistical support.

Additional Information and Declarations

Competing Interests

Author Contributions

The authors declare no competing interests.

T. Mitchell Aide conceived and designed the experiments, performed the experiments, analyzed the data, wrote the paper.

Carlos Corrada-Bravo conceived and designed the experiments, performed the experiments, contributed reagents/materials/analysis tools, wrote the paper.

Marconi Campos-Cerqueira performed the experiments, analyzed the data, wrote the paper.

Carlos Milan, Giovany Vega and Rafael Alvarez performed the experiments, contributed reagents/materials/analysis tools.

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
