# Peer review of "Real-time bioacoustics monitoring and automated species identification"

_PeerJ, doi:10.7717/peerj.103_

## Round 0.1 · original submission · Minor Revisions

All reviewers report the manuscript is interesting and useful. They also provide valuable comments to improve the paper. I suggest the authors fully address the issues raised by the reviewers, especially questions about the experimental design and the validity of your results (e.g. the practicability and efficiency of your monitoring system at a broader taxonomic scale).

·

Basic reporting

I think the manuscript is well written and presents a description of the methodology and first results of an exciting and innovative methodology for real time sound monitoring than will be of use for a good number of research and nature conservation teams around the world and will be of interest to an even wider audience related to bioacoustics and nature conservation.

Experimental design

I think the experimental design described is both innovative and unique and is a breakthrough in terrestrial acoustic monitoring. I strongly believe the authors should expand slightly their description so as to include an evaluation of data loss (maybe at the beginning and at the end of the development of the program). Expected data loss due to malfunction of the different components of the system is important to consider for other researchers or monitors that envision adopting the described technique. Also some sort of economic cost estimate would be a welcome addition to the ms.

Validity of the findings

The results section itself present interesting data on the presence and acoustic activity of a small subset of animals of different taxonomic groups and with different acoustic characteristics. The results are well illustrated and commented. However, the main strength of the work presented in this paper is the methodology presented and the findings presented are of great importance because they show that the system works and is of wide application.

Additional comments

Basic reporting
I think the manuscript is well written and presents a description of the methodology and first results of an exciting and innovative methodology for real time sound monitoring than will be of use for a good number of research and nature conservation teams around the world and will be of interest to an even wider audience related to bioacoustics and nature conservation.

Experimental design
I think the experimental design described is both innovative and unique and is a breakthrough in terrestrial acoustic monitoring. I strongly believe the authors should expand slightly their description so as to include an evaluation of data loss (maybe at the beginning and at the end of the development of the program). Expected data loss due to malfunction of the different components of the system is important to consider for other researchers or monitors that envision adopting the described technique. Also some sort of economic cost estimate would be a welcome addition to the ms.

Validity of the findings
The results section itself present interesting data on the presence and acoustic activity of a small subset of animals of different taxonomic groups and with different acoustic characteristics. The results are well illustrated and commented. However, the main strength of the work presented in this paper is the methodology presented and the findings presented are of great importance because they show that the system works and is of wide application.

Specific comments

Page 3 last paragraph 3rd line
“solve these problems, researchers have developed algorithms to automated species identification”
should be
“solve these problems, researchers have developed algorithms for automated species identification”

page 3 Data acquisition. First paragraph.
“An application in the iPod controls the duration of the recording and the
time between recordings.” Please state how recording gain (level) was set (manual or automatic, and if manual what was the recording level or other calibration procedures). Along those lines if it is possible, it would be interesting to have some idea of the effective area covered by the recording station(s). Although, this, of course, will vary with the source levels of the species studied, perhaps mentioning the effective area (radius) for the focal species with the most intense sound and for the species with the least intense sound… see Llusia et al 2011 (Llusia, D., R. Márquez & R. G. Bowker. 2011. Terrestrial Sound Monitoring Systems, A methodology for quantitative calibration. Bioacoustics 20: 277-286.)
“Presently, it is programmed to record 1 minute of audio every 10
minutes for a total of 144 1-minute recordings per day.” Also here, it would be interesting to know how this recording protocol was decided.

page 4 in data processing,
first paragraph “This spectrogram is created using a short-time Fourier transform (STFT) using 512 samples and a Hann window overlapping 256 samples. For one-minute recordings with a sample rate of 44,100 samples per second each cell of the matrix represents 86 Hz by 0.005 s.” The rationale behind the choice of spectrogram parameters should be explained here. I think it is really interesting methodologically to comment what happens if longer window samples are used.

end of first paragraph
after “The smaller size of mp3 files makes them more appropriate for a web application.”
Maybe a sentence or two should be added here about how the potential loss of information in the MP3 encoding is not relevant (because the original uncompressed files are stored) or has minor effects.


Page 5

“The application is designed so that the user develops the model themselves using tools” should be replaced by (plural of users)
“The application is designed so that the users develop the model themselves using tools”

page 7

second paragraph “but they do not always detect the
species when it is present.” I suggests rewording it “but they do not always detect the species when it is present according to the human expert ear.”

Table 1. footer or table: include sample sizes of training sets

Reviewer 2 ·

Basic reporting

No comments

Experimental design

No comments

Validity of the findings

The technology is useful and interesting. However, I think the authors ignore the limitations and exaggerate the practicability of this technology. This should be revised and clarified in text.

Additional comments

This paper described a technology of bioacoustics monitoring and automated species identification contains hardware and software. The technology is useful and interesting. However, I think the authors ignore the limitations and exaggerate the practicability of this technology.
These limitations should be clarified in text:
1. Many species never produce sound or rarely produce sound or only produce very low intensity sound. For anurans, almost all the females never produce calls. For seasonal reproductive animals, especial for the species with very short reproductive period (only 1-2 weeks), it is difficult to monitor the population dynamics by this technology.
2. For the system, only one microphone was used in a fixed location. For the species with small population and like move frequently in a large scope, it is difficult to record the calls of the animals with a fixed microphone.
3. The sound pressure level of the calls varies largely intraspecies and interspecies. How to set the recording level control? Is it automated regulated or fixed? Do the systems record the sound pressure level?
4. How to filter the serious background noise in field, such as biological noise and abiotic noise?
5. For the very intensity chorus, it is difficult to record the clear single call.
6. For the fixed system, only can record the calls in a relative small scope. The kind of species can monitor use this system is limited.
7. The sound intensity attenuated with the recording distance. Further, for different frequency component of the sound, the attenuation extent is different largely. The system do not record the distance between the calling animal and microphone, so this will result in frequency character distorted with different distance.
8. In method, Data processing, “For one-minute recordings with a sample rate of 44,100 samples per second each cell of the matrix represents 86 Hz by 0.005 s.” Please check the data 0.005s, it should be 0.015s?
9. Please explain the detection frequency in Fig.3 and Fig.4

Reviewer 3 ·

Basic reporting

The ms aim was to describe the acoustical component of the Automated Remote Biodiversity Monitoring Network (ARBIMON), a novel combination of hardware and software for automating data acquisition, data management, and species identification based on audio recordings.
The ARBIMON system seems interesting and important for monitoring fauna.
However, as a review article I missed a literature survey on other automated fauna monitoring systems (describing the vantages to use arbimon instead other automated systems like wildlife acoustics, for instance). The authors also did not include enough information on traditional methods to monitoring fauna and few were associated to clima change. I would like to see more appropriated review in this field.
Please, next time include sequential line numbering.
Intoduction:
2nd. Paragraph:
Comment:
Include reference on: For some economically important species (e.g. salmon) we have long-term data, …-
3rd. Paragraph:
Furthermore, most fauna monitoring protocols require the presence of experts in the field because data are often acquired through indirect cues (e.g. animal vocalizations)…
Comment:
The vocalization is not the unique example of indirect cues to detect fauna. I would like to hear more on other methods to monitoring fauna like transects and mentions on other kind of cues to detect fauna like animals track, feces etc. Would be interesting if you also include the animals’ vocalization vantage and importance. Please, include the usage of this method nowadays with bats, for instance.
Reword the statement and exclude this tiny phrase: This creates various problems.
Thus, you can follow your sentences: …in terms of acoustic identification, there are few experts in comparison to the demand. (include: In addition, the) experts vary in their abilities to correctly identify species, and this leads to observer (observer bias or recording bias?) bias (Fitzpatrick et al., 2009).

Additionally, (include:these protocols) often collect data over a very limited spatial and temporal scale, and these constraints reduce the researcher’s ability to understand the dynamic patterns of animal populations.
So far, you did not mention any protocol. You just said the lack of experts’ availability and abilities to identify species.

Furthermore, there is no permanent record to permit the data verification.
comment:
After Furthermore include: by this traditional method,
I suggest to start a new paragraph from here and exclude- In contrast:
The automated digital recording systems, instead traditional methods, can monitor animal populations 24 hours a day…

This type of monitoring can be effective because in most ecosystems a large proportion of the fauna emits sounds for a variety of reasons including inter and intraspecific communication, orientation (Peter and Slabbekoorn, 2004), and detection and localization of prey and predators (Richardson et al., 1995). Furthermore, these sounds are species specific.
comment:
Avoid small sentences: Furthermore, these sounds are species specific. My suggestion is:
This type of monitoring can be effective because detect species specific sounds. In most ecosystems a large proportion of the fauna emits sounds for a variety of reasons including inter and intraspecific communication, orientation (Peter and Slabbekoorn, 2004), and detection and localization of prey and predators (Richardson et al., 1995).
Methods
Comment:
For all equipment (battery, solar panel, microphone, amplifier, etc), please include the model, city and country of origin).
Results
Comments:
This first paragraph is not part of your results. You must cite this information in methods session. In addition, I would like to know the location (coordinates) of each station in those countries and why does it chosen?
You said that ARBIMON could detect those species (Table 1) in more than 76%. There is any published data on detection of these species to support your system ARBIMON? If yes, please include references.
DISCUSSION
Comments
The first paragraph is not a discussion at all; it seems a conclusion. Please reword it and whole discussion including your data discussion.

Experimental design

The submission fit the journal scope.
As an experimental design the authors showed how does ARBIMON works and present some species examples that they monitored by this system in Costa Rica. Why do the authors choose those species?
Please, clarify the ARBIMON constraints to proceeding validation:
The authors showed that to validate calls there are many occurrences possibilities -negative and false negative/ positive and false positive- I would like to know if there is an expected error (%) for each case? Is there any constraint to do that?

Validity of the findings

Does the species vocalization (used in this article) was already published in other article as monitored fauna? I am asking that, because the authors’ aim was to describe and present a new system to monitoring fauna. However, how do I know if this system was sufficient to detect those animals if we do not have safe data (references or traditional method) to compare and confirm such efficiency?

Additional comments

I do not think that this ms is a review. I am missing a literature survey on other automated systems and also mention on traditional methods to monitoring fauna. I would like to see more appropriated review in this field.

---

## Round 0.2 · Minor Revisions

Thanks for addressing the issues raised by the three reviewers. I am happy to accept the manuscript after the following comment is addressed. The authors replied the reviewer 3's question "Why do the authors choose those species" as "The species were selected to cover a range of taxa with different types of vocalizations.", and presented species-specific identification models for these species, including two so-called "unidentified insect (insect #1, #2)". The authors should at least make some explanations about why the two insect species cannot be identified, in the final version of your paper.

---

## Round 0.3 · accepted · Accept

Thanks for addressing the comment about species identification. I am happy to accept the manuscript.